# Effects of Plasma-Activated Water on Skin Wound Healing in Mice

**DOI:** 10.3390/microorganisms8071091

**Published:** 2020-07-21

**Authors:** Dehui Xu, Shuai Wang, Bing Li, Miao Qi, Rui Feng, Qiaosong Li, Hao Zhang, Hailan Chen, Michael G Kong

**Affiliations:** 1State Key Laboratory of Electrical Insulation and Power Equipment, Centre for Plasma Biomedicine, Xi’an Jiaotong University, Xi’an 710049, China; fr19960312@stu.xjtu.edu.cn (R.F.); liqiaosong0318@163.com (Q.L.); zhang216@mail.xjtu.edu.cn (H.Z.); 2The school of Life Science and Technology, Xi’an Jiaotong University, Xi’an 710049, China; shuaiwang3059@stu.xjtu.edu.cn (S.W.); lb905533242@stu.xjtu.edu.cn (B.L.); camille77@163.com (M.Q.); 3Frank Reidy Center for Bioelectrics, Old Dominion University, Norfolk, VA 23508, USA; h1chen@odu.edu; 4Department of Electrical and Computer Engineering, Old Dominion University, Norfolk, VA 23529, USA

**Keywords:** plasma-activated water, bacterial inactivation, wound healing

## Abstract

Cold atmospheric plasma (CAP) has been widely used in biomedicine during the last two decades. While direct plasma treatment has been reported to promote wound healing, its application can be uneven and inconvenient. In this study, we first activated water with a portable dielectric barrier discharge plasma device and evaluated the inactivation effect of plasma-activated water (PAW) on several kinds of bacteria that commonly infect wounds. The results show that PAW can effectively inactivate these bacteria. Then, we activated tap water and examined the efficacy of PAW on wound healing in a mouse model of full-thickness skin wounds. We found that wound healing in mice treated with PAW was significantly faster compared with the control group. Histological analysis of the skin tissue of mice wounds showed a significant reduction in the number of inflammatory cells in the PAW treatment group. To identify the possible mechanism by which PAW promotes wound healing, we analyzed changes in the profiles of wound bacteria after PAW treatment. The results show that PAW can significantly reduce the abundance of wound bacteria in the treatment group. The results of biochemical blood tests and histological analysis of major internal organs in the mice show that PAW had no obvious side effects. Taken together, these results indicate that PAW may be a new and effective method for promoting wound healing without side effects.

## 1. Introduction

Wounds can be caused in various ways, such as through blunt instrument injuries, scrapes, falls, and knife injuries. Wound healing in humans is an extremely complex process that is triggered immediately after injury through the activation of numerous biological pathways in the body [1,2]. In most cases, wounds become infected by various microorganisms, which can lead to serious inflammation and significantly delay the wound healing process. In some extreme cases, serious microbial infection of wounds can lead to irreparable damage, potentially placing a serious financial and mental burden on patients and seriously affecting their quality of life [3]. Clinically, skin wounds are usually treated through debridement, anti-inflammatories, and other types of treatment to prevent infection and promote wound healing. Additionally, antibiotics are often used to assist with repressing inflammation caused by bacterial infection and promote healing. However, the abuse of antibiotics can produce side effects and contribute to the increasing emergence of antibiotic resistance [4]. Therefore, new technologies must be found that can effectively promote wound healing while being low cost, easy to use, and with less side effects.

Cold atmospheric plasma (CAP) is a partly ionized gas composed of a large number of charged particles (such as electrons and ions), free radicals, metastable molecules, electrically neutral atoms, and ultraviolet radiation [5]. CAP can be generated without requiring a vacuum environment [6], and the apparent temperature of CAP is close to room temperature, making it possible to directly treat the cells or tissues without causing thermal damage [7,8,9]. CAP can be used to generate reactive oxygen species (ROS) and reactive nitrogen species (RNS), including H_2_O_2_, NO_2_^−^, NO_3_^−^, O_3_, ONOO^−^, ^•^OH, and O_2_^•−^ [10,11,12,13,14], which play important roles in biomedical applications [15,16], such as cancer therapy [17,18,19], hemostasis [20], wound healing [21,22,23], and skin disease [24]. CAP is highly efficient for inactivation of different kinds of bacteria such as Gram-positive, Gram-negative bacteria and fungus [25,26], as well as the drug-resistant microorganisms and bacterial biofilms [27,28], which is benefit for the bacterial infected wound. During wound healing, keratinocytes, fibroblasts, and endothelial cells are the main cell populations involved in re-epithelialization and the construction of granulated tissue [29]; thus, the ability of these cells to migrate and proliferate plays an important role in the wound healing process. Studies have shown that CAP can improve the viability, migration capacity, and reproduction capacity of keratinocytes and fibroblasts [30,31,32]. In addition, CAP has been shown to accelerate wound healing through recruitment of neutrophils, production of growth factors, and promotion of angiogenesis [33,34,35]. Therefore, CAP has a promising future in the application of skin wound healing [36].

Currently, most plasma jet devices and dielectric barrier discharge (DBD) plasma devices are supplied by a large-sized high-voltage power supply and a large gas cylinder [11,37,38,39], which limits application in laboratories. For animal experiments, the current gas plasma system is often too large to be implemented in animal experiment centers for direct plasma treatment. However, researchers have reported that plasma-activated water (PAW), which also contains various ROS and RNS [40,41], produces similar biological effects to direct plasma treatment. A time delay exists between the preparation of PAW in the laboratory and the transfer to the animal center for plasma treatment of animals. This would lead to a decrease in ROS and RNS in PAW, especially for short-lived species and, hence, a reduction in the therapeutic biological effects of PAW treatment. Hence, we developed a small and portable DBD air plasma device (300 mm × 250 mm × 30 mm) that is simple and can conveniently be transported to animal centers for immediate preparation of PAW and treatment of animals. We evaluated the antibacterial effects of the immediate preparation of PAW, examining the effect produced on the healing of skin wounds in mice. In addition, we investigated the flora composition of the wound site using DNA sequencing for a detailed analysis of the in vivo inactivation of various bacteria. Lastly, we examined whether PAW treatment had any side effects compared to the control group.

## 2. Materials and Methods

### 2.1. Animals and Wound Creation

Male Institute of Cancer Research (ICR) mice (Specific Pathogen Free, 30 ± 5 g) were purchased from and raised in the standard breeding environment in the Animal Experiment Center of Xi’an Jiaotong University (Xi’an, China). In this study, all experiments were designed and operated in accordance with animal welfare guidelines with Ethic Committee approval (No. xjtu2018-215). During the construction of the wound model, each mouse was injected with an appropriate amount of 1% pentobarbital sodium (an anesthetic) according to body weight. After the mice were anesthetized, the hair on the back of each mouse was removed using depilatory cream, and the hair removal site was wiped with 75% alcohol. We then used sterile medical scissors to cut a round wound with a diameter of 2 cm on the back of each mouse, effectively removing the skin down to the muscular layer including the dermis and epidermis. Then, a 200 μL (5 × 10^6^/mL) suspension of *Pseudomonas aeruginosa* was added to the wound, and a piece of sterile gauze was then fixed to the wound. After the wounds had been infected (approximately 3 days after wound cutting), 10 mice were randomly divided into a control group and a treatment group.

### 2.2. Device Introduction and Wound Treatment

In this study, we used a portable DBD plasma device that was developed by the Center for Plasma Biomedicine at Xi’an Jiaotong University. A schematic diagram of the internal connection of the device is shown in Figure 1. The whole device is mainly composed of an alternating current (AC) booster system, direct current (DC) output system, plasma generator, cooling fan, and air pump. The gas flow rate provided by the air pump was about 15 L/min. The DBD structure of the plasma consisted of a high-voltage (HV) electrode, a ground electrode, and a 1 mm thick hexagonal polytetrafluroethylene (PTFE) component sandwiched between the two electrodes. The DBD plasma was generated when the device was connected to 220 V AC. The output of the AC boost system was standard sinusoidal voltage with a peak-to-peak value of 6 kV and at a frequency of 15 kHz. DBD plasma was maintained at an electrical power of 3.3 W. This small and portable DBD air plasma device could be conveniently transported to the animal center for immediate preparation of PAW and following plasma treatment on the animals.

Following the infection of skin wounds, 100 mL of tap water was activated for 5 min each time before being used to treat the infected wounds of mice in the treatment group. Treatment was carried out by applying 10 mL of PAW to the wound area of mice in the treatment group, with an equal amount of nonactivated tap water used in the control group. The timeline for the entire animal experiment is shown in Figure 4.

### 2.3. Wound Healing Analysis

During wound treatment with PAW, the wound healing of mice in the control and treatment groups was photographed at different time points. A ruler was placed next to each mouse when pictures were taken, with the camera placed at the same height every time. The time required for the wound to heal for each of the mice in the treatment and control groups was recorded. All operations were performed by the same experimenter to avoid additional errors due to variation.

### 2.4. Histological Analysis of Wound Skin

On day 10, a small wound skin sample (2 mm × 2 mm) of each group was obtained by biopsy and fixed with 4% polyphenol formaldehyde. The samples of wound skin tissue were stained and photographed by Wuhan Seville Biotechnology Company (Wuhan, China).

### 2.5. Inactivation of Bacteria by PAW

The antibacterial efficiency of PAW was evaluated by total bacterial Adenosine Triphosphate (ATP) measurement using an ATP detector (TIANLONG Co., Xi’an, China) according to the manufacturer’s instructions, and the relative bacterial number as detected with a high-throughput flow cytometry (Accuri C6, BD, Franklin Lakes, NJ, USA). *Pseudomonas aeruginosa* was used as it is the most common bacteria found in infected wounds. For inactivation experiments, 0.5 mL of 5 × 10^7^/mL *Pseudomonas aeruginosa* was centrifuged at 4600 rpm for 4 min and the supernatant discarded. Then, 1 mL PAW (with different activating times) was incubated for a set period of time in the PAW treatment group, with nonactivated water being used instead in the control group. After treatment, the bacteria were resuspended in 5 mL culture medium (Luria-Bertani Medium) for proliferation in a shaking machine for 1–2 h. When the concentration in the control group reached approximately 5 × 10^7^/mL (as detected by flow cytometry), the bacteria from both groups were obtained and analyzed using an ATP detector and flow cytometry for ATP activity and relative bacteria number, respectively. We also investigated the in vitro antibacterial effects of PAW on other bacteria (*Escherichia coli*, *Salmonella paratyphi-B*, and *Staphylococcus aureus*), in addition to *Pseudomonas aeruginosa*, using a colony counting assay as conducted by a third-party organization (Shaanxi Institute of Microbiology, Shaanxi, China). Briefly, samples were serially diluted, and 10 µL of each dilution was spotted onto different agar plates and incubated overnight at 37 °C. Among them, Luria-Bertani agar was used to culture *Pseudomonas aeruginosa*, *Escherichia coli*, *Staphylococcus aureus* and Buffer Peptone Water agar was used to culture *Salmonella paratyphi*-*B*. The numbers of surviving bacteria were determined by counting the colonies (marked in blue dots).

### 2.6. Analysis of Bacterial Inactivation in Wounds by DNA Sequencing

On day 10, bacterial samples from wounds were obtained by repeatedly wiping the wounds of mice in the control group and the PAW treatment group with wet cotton swabs. The bacterial samples on the wound were sent to Shanghai Biotree Biotechnology Company (Shanghai, China) for DNA sequencing. The obtained sequence information was used for flora composition analysis toward systematically analyzing the type and relative abundance of bacteria in the wound after PAW treatment to allow assessment of bacterial inactivation in the wound after plasma treatment.

### 2.7. Analysis of Biological Safety of PAW

On day 23, the mice from both groups were injected with 1% pentobarbital sodium. Blood was collected from the hearts of comatose mice using a syringe, and placed in a 1.5 mL centrifuge tube at room temperature for 2 h prior to centrifugation at 3000 rpm for 15 min to collect the serum for detection of biochemical indicators, including liver function, kidney function, blood lipids, blood glucose, inorganic ions, or antioxidants. The mice were then sacrificed, and the main internal organs (heart, liver, spleen, lung, and kidney) were dissected and fixed with 4% paraformaldehyde. The serum samples and the visceral tissue samples were respectively tested and stained by Wuhan Seville Biotechnology Company (Wuhan, China).

### 2.8. Statistical Analysis

The data were processed using GraphPad Prism 5.01 (GraphPad Software, San Diego, Calif, USA) statistical software. All experimental results are presented as the mean ± standard deviation (SD) of at least three independent experiments. Student’s *t*-test was applied to evaluate statistical significance. Values of *p* < 0.05 between two independent groups was considered to be statistically significant.

## 3. Results and Discussion

### 3.1. Inactivation of Several Kinds of Bacteria That Commonly Infect Wounds

An important factor that hinders wound healing is the susceptibility of wounds to bacterial infection. We evaluated the inactivation effect of PAW on *Pseudomonas aeruginosa*, which is one of the most common bacteria found in infected wounds. The results of this experiment used to assess the effect of PAW in inactivating *Pseudomonas aeruginosa* are shown in Figure 2. Figure 2A,B show the bacterial inactivation efficiency following 5 min of incubation with PAW that had been activated for 1, 3, and 5 min, as assessed using *Pseudomonas aeruginosa*. The sterilization effect of PAW on bacteria was also evaluated by measuring the total ATP content in bacteria and the relative total amount of bacteria. Figure 2C,D show the results of activating water for 5 min and the effects of PAW incubation for 3, 7, and 11 min on *Pseudomonas aeruginosa*. The sterilization effect of PAW on bacteria was evaluated by measuring the total bacterial ATP content using an ATP detector and the relative total amount of bacteria using flow cytometry. Figure 2A,B show that the sterilization effect of the PAW was not ideal when activating water for 1 and 3 min. However, when the activation time was increased to 5 min, PAW was able to inactivate most of the *Pseudomonas aeruginosa* within 5 min. Figure 2C,D show that the measurements of both the total ATP in the bacteria and the relative total bacterial count indicate that the total amount of bacteria decreased as PAW incubation time increased. When activating water for 5 min, PAW was found to inactivate most of the *Pseudomonas aeruginosa* within 3 min, and the sterilization effect was found to increase as incubation time increased.

The results of Figure 2 reveal the antibacterial effects of PAW against *Pseudomonas aeruginosa*. This antibacterial effect was plasma-dose-dependent. The sterilization effect of PAW was incomplete when activating water for 1 or 3 min, but was effective with 5 min (Figure 2A,B). The ROS concentration in the PAW increases with the activation time, and CAP treatment destroys the integrity of the bacterial membrane [42], which may be due to the concentration of active particles in the PAW not being high enough to inactivate *Pseudomonas aeruginosa* when only using activation times of 1 and 3 min. The sterilization effect produced by activating water for 5 min was almost the same as the results after activating water for 10 min [42]. Hence, the sterilization effect can be enhanced by appropriately increasing the activation time. According to the results in Figure 2, we conclude that activation time and incubation time are two important factors for inactivating bacteria, so a high sterilization effect can be achieved in a relatively short time by selecting an appropriate activation time and incubation time. Because Figure 2B,D represent different groups of the experiments, errors will occur when comparing the experimental results for different groups. For example, the relative quantity of *Pseudomonas aeruginosa* from the 5-min incubation using PAW that has been activated for five minutes was higher than the relative quantity of *Pseudomonas aeruginosa* for the 3-min incubation with PAW that had been activated for 5 min.

In addition, we confirmed the in vitro antibacterial effects of PAW through a colony counting assay, carried out by a third-party organization, in which inactivation of *Pseudomonas aeruginosa*, *Escherichia coli*, *Salmonella paratyphi*-*B*, and *Staphylococcus aureus* were assessed [43]. The results of colony counting assay showed that 5 min of PAW activation could efficiently inactivate all assayed bacteria (Figure 3A–D). Figure 3E shows the statistical results of the sterilization effect. Figure 3A–E shows that compared with the control group, *Pseudomonas aeruginosa*, *Escherichia coli*, *Salmonella paratyphi*-*B*, and *Staphylococcus aureus* did not form obvious colonies after treatment with PAW, even in the undiluted samples. Treatment with PAW that had been activated for 5 min resulted in the reduced of bacterial count by more than 99.9999%. Our results indicate improvement of the inactivation efficacy compared to results reported in the literature for *Escherichia coli*, in which 15–30 min were required to achieve the same results [44,45].

### 3.2. Skin Wounds Healing in Mice

During the wound healing process, the two groups of mice were kept in the same breeding environment and had access to sufficient food and water. The mice in the treatment group and the mice in the control group were treated with PAW and normal tap water, respectively. The timeline for the whole animal experiments is shown in Figure 4. The results of wound healing are shown in Figure 5. To assess healing, photographs of the wounds in mice from the PAW treatment and control groups were taken at different time points (days 0, 4, 7, 10, and 17) (Figure 5A,B). The results of the statistical analysis of the diameter of wounds in mice from the treatment and control groups on days 0, 4, 7, 10, and 17 are shown in Figure 5C. A histogram of the statistical results of the time required for complete wound healing in the two groups of mice is shown in Figure 5D. As shown in Figure 5A,B, the wounds of mice from both groups decreased over time, and the wounds of mice from the treatment group were obviously smaller than those of the control group. On day 17, no obvious wounds were observed in the mice treated with PAW, but we observed that the wounds on mice from the control group were not completely healed. This indicates that PAW can promote skin wound healing in mice. As shown Figure 5C, the wound diameters of mice in the control and treatment groups all decreased over time; however, the diameters of the wounds in mice from the PAW treatment group (denoted by red lines) decreased significantly faster than in the control group (denoted by blue lines). Figure 5D shows that the time required for the wounds to completely heal in the treatment group was significantly shorter than for the control group. These results show that PAW can effectively promote the healing of skin wounds in mice.

Wound healing in humans is an extremely complex process [1]. Many key factors affect the process of wound healing, including the host (wound), the bacteria, and treatment parameters [36]. In this study, a wound infection model was used to evaluate the effect of PAW on wound healing in mice. According to the results in Figure 5A,B, on day 7, no obvious pus was found in the wounds of mice in the treatment group, but the control group mice still had pus on their wounds on days 7 and 10. Hence, PAW treatment of mice wounds appears to inhibit the growth of bacteria. The skin is mainly composed of keratinocytes and fibroblasts, and these two cell types play an important role in the wound healing process. In vitro experiments have shown that direct plasma treatment can enhance the migration ability of keratinocytes and fibroblasts [32]. Therefore, we speculated that PAW promotes wound healing for two main reasons: PAW markedly reduces the bacterial burden on the wound, which plays a critical role in promoting wound healing by PAW treatment; PAW may also enhance the migration ability of keratinocytes and fibroblasts in the skin around the wound. Here, complete healing means that there was no obvious wound on the back of the mouse, such as on day 17 in Figure 5B.

The results of H&E staining of the skin wounds in mice from the two groups are shown in Figure 6. The results show that the number of inflammatory cells (indicated with black arrows) in the skin wound tissue of the PAW treatment group was significantly lower than in the control group. This result is consistent with the results in Figure 5. PAW treatment reduces the burden of bacteria on the wound considerably resulting in less severe inflammatory reaction in the tissue, thus promoting the wound healing process.

### 3.3. Analysis of In Vivo Bacterial Inactivation by DNA Sequencing

We further analyzed the in vivo bacterial inactivation using DNA sequencing on the wound site after PAW treatment. Although the wound was infected using *Pseudomonas aeruginosa*, other bacteria in the environment could potentially contaminate the wound and result in polybacterial infection. The Wayne diagram in Figure 7 shows that the types of bacterial operational taxonomic units (OTU) present significantly differed between the control group and the PAW treatment group. A cluster analysis heat map was used to determine the differences in the abundance of bacterial genera on the wound site (Figure 8). We found that for most bacterial genera after PAW treatment, their abundance was reduced compared to the control. By calculating the most abundant bacterial OTU in the sample, we found that bacteria of *Azoarcus* (Bacteria, Proteobacteria, Betaproteobacteria, Rhodocyclales, Rhodocyclaceae), *Enterococcus* (Bacteria, Firmicutes, Bacilli, Lactobacillales, Enterococcaceae); TM7_3 (Bacteria, TM7), and Spirochaetes (Bacteria, Spirochaetes) were significantly lower in abundance after PAW treatment. Figure 9 depicts the average and median values of the relative abundance of each genus. These results provided in vivo confirmation that PAW treatment could reduce bacterial burden at the wound site and promote wound healing.

### 3.4. Analysis of Biological Safety

Because PAW must be in direct contact with wounds to promote healing, biosafety is an important factor to consider. In a previous experiment, we demonstrated the effect of PAW on the healing of mice skin wounds. Here, we evaluated the safety of using PAW on the skin tissues of mice using blood biochemical indicators and a histological analysis of major internal organs. The results are shown in Table 1 and Figure 10. The blood biochemical indicators and the internal organ tissue structure of the mice in the PAW treatment group were not significantly different from those of the mice in the control group, indicating this treatment did not detrimentally affect organ function.

According to the results in Table 1, PAW had no significant effect on liver function (albumin and aspartate aminotransferase), kidney function (urea nitrogen), blood lipids (total cholesterol), blood glucose (glucose), inorganic ions (potassium), or antioxidants (total superoxide dismutase) in mice. Figure 10 shows the results of histopathological sections after H&E staining. H&E staining can show changes in tissue structure and cell composition. The staining results in Figure 10 show that there was no obvious abnormal change in the tissue structure and cell composition of the mice after PAW treatment compared with the control group. From the above results, we conclude that PAW is safe for mice. Several groups also reported that treatment with plasma-activated medium is safe on mice with no significant changes in the assayed indicators [46,47].

## 4. Conclusions

In this study, we evaluated the use of PAW in treatment of skin wounds in male ICR mice, in which we created full-thickness skin wounds that were mainly infected with *Pseudomonas aeruginosa*. For the treatment of skin wounds, we used a portable DBD plasma device to activate tap water and generate PAW. We demonstrated that immediate preparation of PAW using our portable DBD device can effectively inactivate a variety of bacteria (*Pseudomonas aeruginosa*, *Escherichia coli*, *Salmonella paratyphi*-*B*, and *Staphylococcus aureus*) as demonstrated in vitro. In vivo experiments of wound infection based on *Pseudomonas aeruginosa* revealed that PAW treatment could accelerate the healing of skin wounds in mice, with a faster healing time observed when compared to the control. By in vivo detection of the bacterial burden on the wound site, we found that PAW treatment significantly reduces the bacteria abundance (especially for *Pseudomonas aeruginosa*), which may play a critical role in promoting wound healing by PAW treatment. In addition, PAW treatment was safe for mice, with no significant changes in major organs, tissue structure, and some important blood biochemical indexes. Our results indicated that immediate PAW treatment with a portable device could be a simple, effective, and cheap method for wound healing in the future.

## Figures and Tables

**Figure 1 microorganisms-08-01091-f001:**
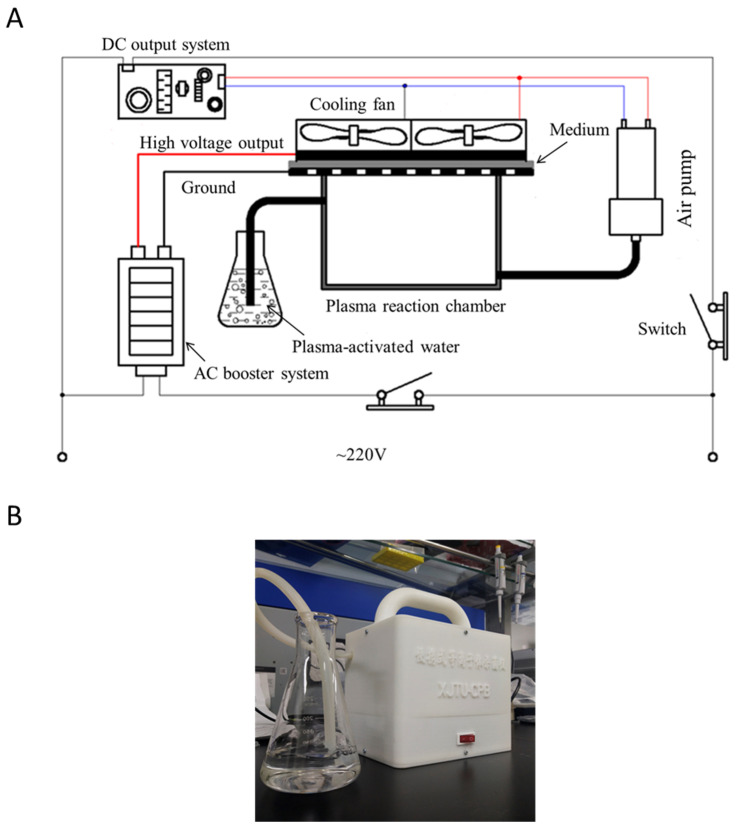
(**A**) Schematic diagram of the internal connection of the portable device and (**B**) photograph of the portable device.

**Figure 2 microorganisms-08-01091-f002:**
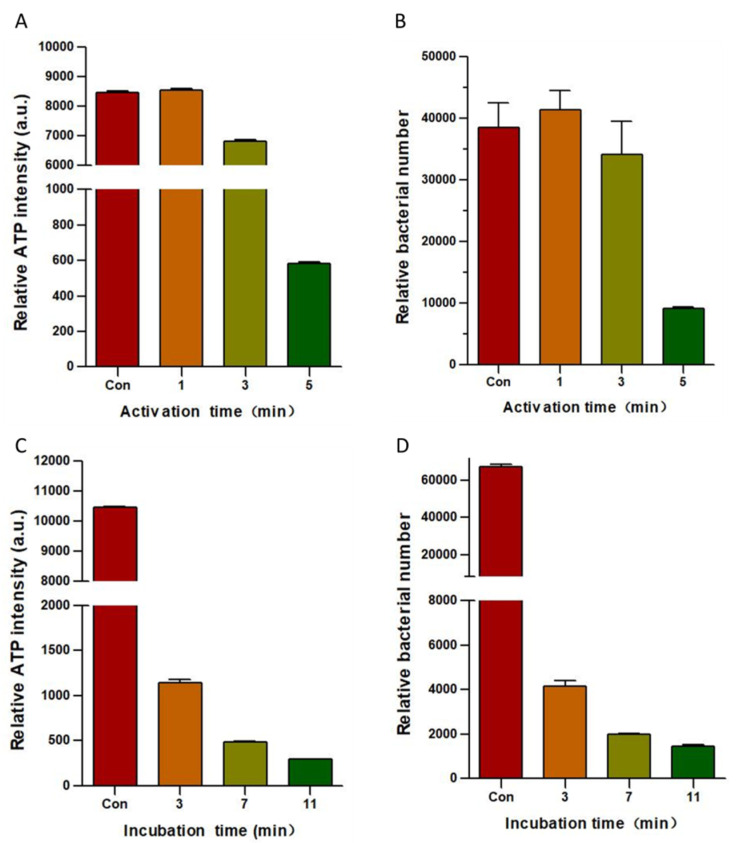
Inactivation effect of plasma-activated water (PAW) on *Pseudomonas aeruginosa*. The bacterial inactivation efficiency on *Pseudomonas aeruginosa* following incubation with PAW for 5 min after activating water for 1, 3, and 5 min in terms of (**A**) relative total ATP content and (**B**) relative total bacterial count. The results of activating water for 5 min and the effects of PAW incubation for 3, 7, and 11 min in terms of (**C**) relative total ATP content and (**D**) relative total bacterial count of *Pseudomonas aeruginosa*. All data are expressed as the mean ± SD of three separate experiments.

**Figure 3 microorganisms-08-01091-f003:**
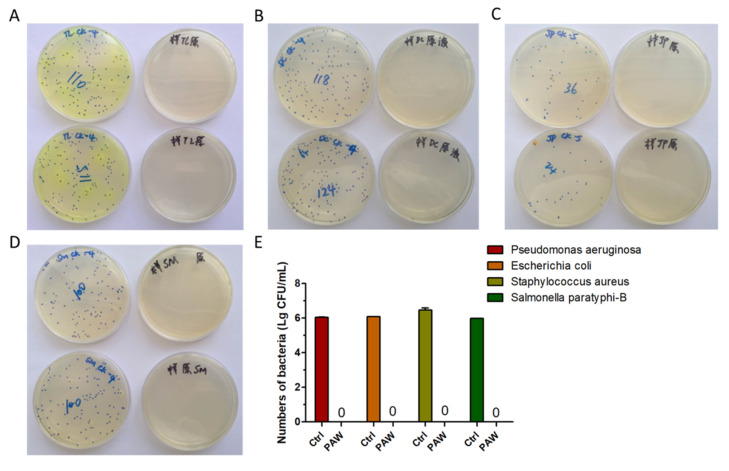
Inactivation effect of PAW on various kinds of bacteria. The results of activating water for 5 min and the effects of PAW incubating on bacteria for 5 min in (**A**) *Pseudomonas aeruginosa*, (**B**) *Escherichia coli*, (**C**) *Staphylococcus aureus*, and (**D**) *Salmonella paratyphi*-*B*, and (**E**) the statistical results of (A–D).

**Figure 4 microorganisms-08-01091-f004:**
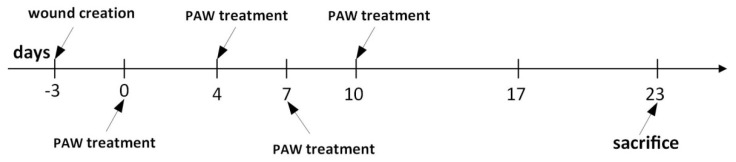
Timeline for animal experiments. The wounds of the mice in the treatment group had completely healed on day 17, and the wounds of the mice in the control group had completely healed on day 23.

**Figure 5 microorganisms-08-01091-f005:**
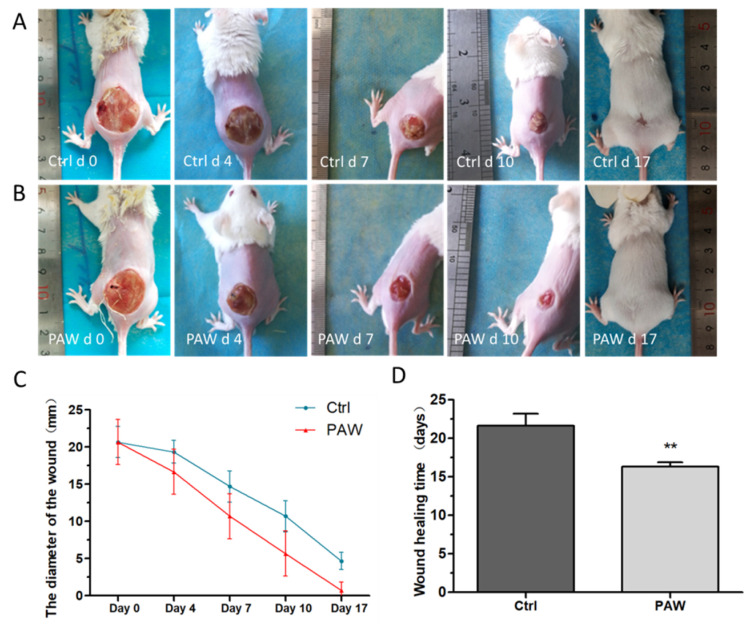
Photographs of the wound healing process. Photographs of the mice wounds were captured at different time points (days 0, 4, 7, 10, and 17) for (**A**) the PAW treatment group and (**B**) the control group. (**C**) The results of the statistical analysis of wound diameters of mice in the treatment group (red lines) and the control group (blue lines) on days 0, 4, 7, 10, and 17. (**D**) The results of the analysis of the time required for the wounds to completely heal in the two groups of mice. Data are expressed as the mean ± SD, *n* = 5; ***p* < 0.01 (Student’s *t*-test).

**Figure 6 microorganisms-08-01091-f006:**
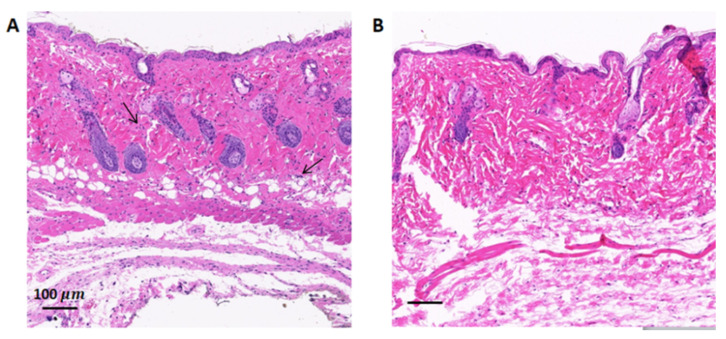
Representative histology of skin wounds in mice on day 10. The wound skin tissues of mice from (**A**) the control group and (**B**) the PAW treatment group were stained by hematoxylin and eosin.

**Figure 7 microorganisms-08-01091-f007:**
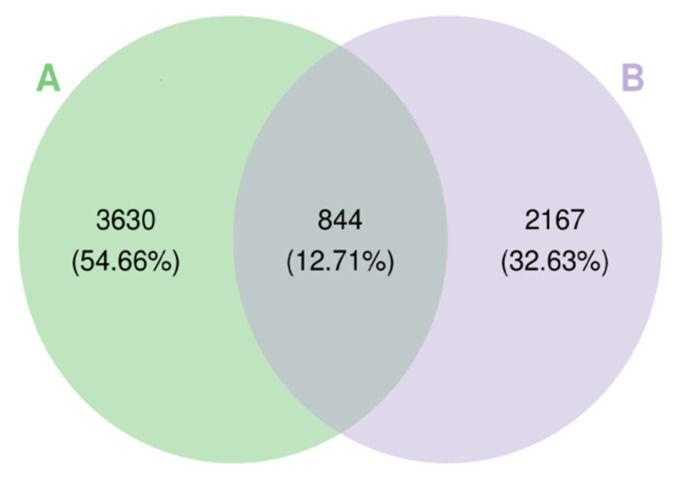
Differences in the types of bacterial operational taxonomic units (OTU) on skin wounds from (**A**) the control and (**B**) PAW treatment groups.

**Figure 8 microorganisms-08-01091-f008:**
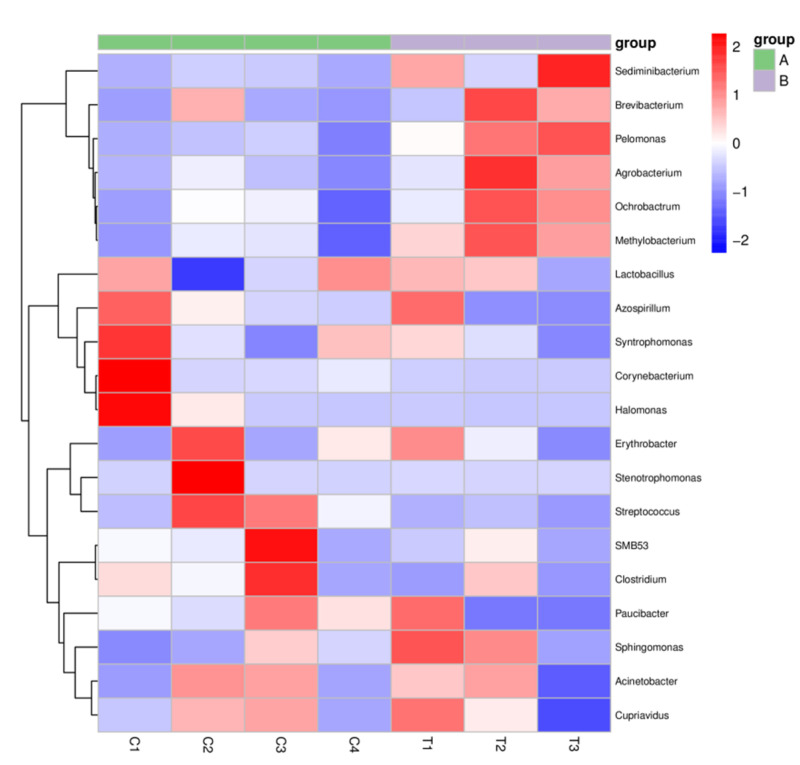
A heat map showing the differences in the bacterial abundance in (**A**) the control (*n* = 4) and (**B**) PAW treatment (*n* = 3) groups. Red indicates that the genus is more abundant in the sample than in other samples, and blue indicates that the genus is less abundant in the sample than other samples.

**Figure 9 microorganisms-08-01091-f009:**
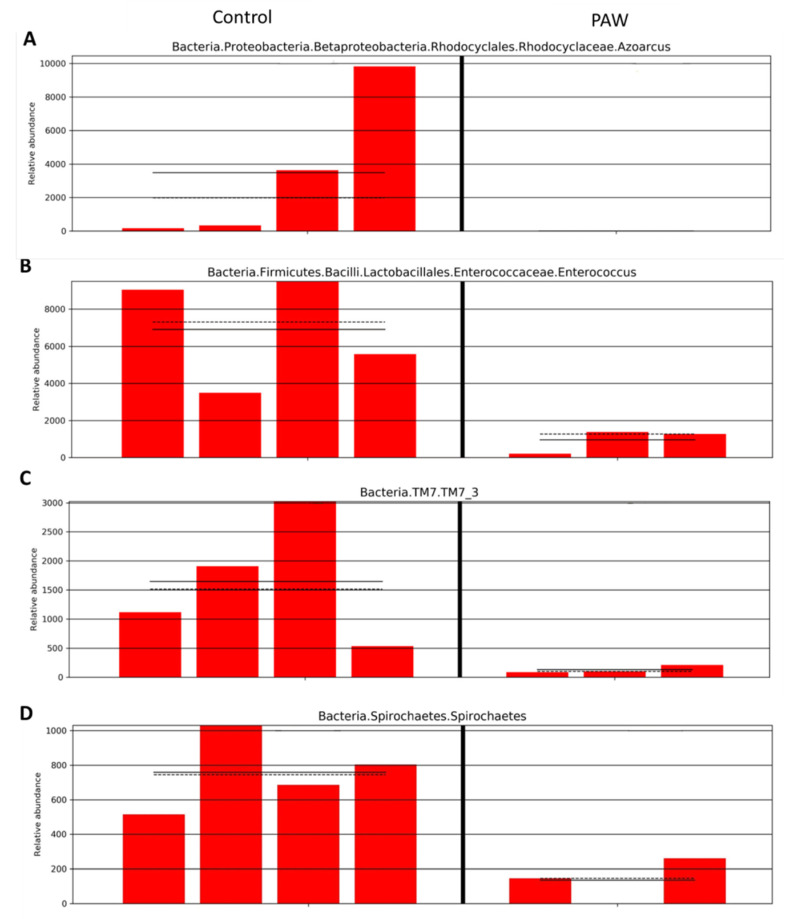
Relative abundance of some major bacterial operational taxonomic units (OTU) in the wound site in the control and PAW treatment group. (**A**) Azoarcus (Bacteria, Proteobacteria, Betaproteobacteria, Rhodocyclales, Rhodocyclaceae), (**B**) Enterococcus (Bacteria, Firmicutes, Bacilli, Lactobacillales, Enterococcaceae), (**C**) TM7_3 (Bacteria, TM7) and (**D**) Spirochaetes (Bacteria, Spirochaetes).

**Figure 10 microorganisms-08-01091-f010:**
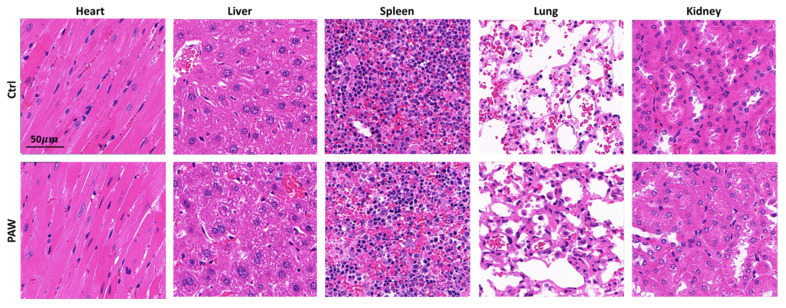
Representative histology of heart, liver, spleen, lung, and kidney stained by hematoxylin and eosin.

**Table 1 microorganisms-08-01091-t001:** Blood biochemical indicators.

Indicator	Control	PAW
Albumin (g/L)	27.21 ± 1.50	26.39 ± 0.96
Aspartate aminotransferase (U/L)	240.66 ± 48.69	245.03 ± 14.80
Urea nitrogen (mg/dL)	23.66 ± 1.70	22.82 ± 1.50
Total cholesterol (mmol/L)	3.84 ± 0.63	4.10 ± 0.51
Glucose (mmol/L)	7.74 ± 1.93	9.47 ± 0.93
Potassium (mmol/L)	11.18 ± 1.19	10.77 ± 1.01
Total superoxide dismutase (U/mL)	323.48 ± 50.91	310.34 ± 47.62

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
