# Peer review of "Effects of Plasma-Activated Water on Skin Wound Healing in Mice"

_microorganisms, 2020, doi:10.3390/microorganisms8071091_

Round 1
Reviewer 1 Report
The manuscript has details comments. My general recommendations are:
1) thorough revise for English presentations in terms of flow, grammar, transitions, etc.
2) revise the scientific presentation of the studied bacteria.
3) a detailed clear description of the protocols is needed.
4) Precise presentation of the results and bridge them to discussion is highly recommended.
5) check list of references & their citation in the list.

Reviewer 2 Report
Even though this is an interesting study, the presentation and explanation of results are very poor and needs significant rewriting explaining each point.
1.
We evaluated the inactivation effect of PAW on several kinds of bacteria that commonly infect wounds using an experimental approach.
But in Fig.2 the authors only used one bacterial strain, Pseudomonas. It is recommended that the authors should also present the results from another bacterial strain to compare their findings.
Experimental methodology corresponding to Fig.2 should be written clearly in detail. Not understandable to the reader. English language editing is necessary. Also results need to be rewritten corresponding to this part. (3.1)
Figure 2C, D shows the sterilization effect of PAW on bacteria
As per Fig. 2B, after 5 min activation of PAW, it shows approximately 10,000 bacterial count after treatment for 5 min. If this is true, in Fig. 2D, the same 5 min activated PAW should have higher number of bacteria after treatment with only 3 min. But here it shows even lesser bacterial count than the count shown in Fig. 2B after 5 min. The authors need to explain this discrepancy.
Round 2
Reviewer 1 Report
Dear authors,
We thank you for the taking into consideration most of the comments, however, there is till alot of work that needs to be done to have a good quality of publication. Please revise carefully our comments on the current and previous manuscripts, as well as on your response letter. Unfortunately, because of the excessive comments we have, our review did not continue beyond line 172, though there are a couple of comments after that as those were captured on a glance.
